## COMMENT

# Disconnected psychology and neuroscience—implications for scientific progress, replicability and the role of publishing

Christian Beste [1,2,3]✉

There has been a fascination for centuries surrounding drivers of human behavior and the relationship between the 'mind' and the brain. However, there is an ongoing disconnection between different research communities aiming to provide a mechanistic understanding about what underlies behavior, psychology and neuroscience. This comment outlines why this is a problem for scientific progress and replicability in brain sciences and considers how publishers can play a central role to help overcome the disconnect between, what should be, joint scientific communities.

There is a strong and enduring fascination about what drives behavior, including mind-brain relationships. Particularly with the advent of experimental scientific psychology, much progress has been made in understanding cognitive, affective, and social processes. This success has been possible because psychological science is firmly rooted in theories and theory-derived hypotheses that can then be tested. However, even nowadays, many influential psychological theories do not directly address the level of neural correlates and processes. This makes it difficult for another ever-growing community of researchers to connect to these theories, i.e., neuroscientists. Consequently, it is challenging to use the conceptual stringency of psychological theories in neuroscience. Conversely, neuroscientists often do not explicitly address relevant psychological theories for the processes that they are investigating at a neural level. Thus, there is (still) an "interface problem" between psychological science and neuroscience, leaving the fields disconnected even though they are making efforts to often address the same questions. This problem is particularly intense when presenting results from studies reporting primary experimental/empirical research in neuroscience and psychology. There are many instances in which there are overlaps between psychological science theories and neuroscience. However, there is an overt segregation between the fields, the degree of which can very between countries and their academic research traditions and practices. Clearly, fields can make progress without considering the other, and it is not necessary to understand, say, the neural level to make significant progress at the behavioral level and vice versa. Philosophers like D. Dennett have long been arguing that there are separate levels of analyzing the mind (e.g., the behavioral level, the neural level etc.), but more holistic or unified understandings of brain function are still impossible with a separation of psychology and neuroscience. Universally, the core problem is still that "perspective-taking" and communication between the above-mentioned communities are often not that widespread and developed to the level required to enable theoretical accounts that unite the communities in a

[1] Cognitive Neurophysiology, Department of Child and Adolescent Psychiatry, Faculty of Medicine, TU Dresden, Dresden, Germany. [2] Cognitive Psychology, Faculty of Psychology, Shandong Normal University, Jinan, China. [3] University Neuropsychology Center, Faculty of Medicine, TU Dresden, Dresden, Germany. ✉email: christian.beste@uniklinikum-dresden.de

sustainable manner. Crucially, this problem impedes both scientific progress and has substantial repercussions in terms of replicability in brain sciences. However, I think that publishers can play a central role in overcoming these problems. Here I provide some general ideas intended to stimulate thinking and discourse about how this may be accomplished.

## Fragmentation of disciplines

As science progresses, a general phenomenon is that different scientific disciplines become more and more fragmented. On a publishing level, this is reflected by an ever-increasing number of specialty journals. Fragmentation in science is typical because knowledge becomes more and specific and fine-grained. Highly-focused specialization in science is therefore the logical and evolutionary consequence of scientific progress. Regarding the connection between psychology and neuroscience, there are often different "grain sizes", or levels, in the consideration of mental processes that may become difficult to integrate. Specifically, psychological theories, which are based predominantly on behavioral data, vary in their specificity and breadth of neural processes upon which these theories can be applied. In the case of neuroscience theories, a variety of methods address different levels of inspection—from the single cell to circuitry to systems. In addition, the methods will vary with respect to temporal and spatial resolution. At present, the difficulties faced when connecting neuroscience research and psychological theories increase when neuroscience research focuses on increasingly smaller building blocks in neuronal mechanisms (e.g., microcircuits, multi-unit, or single-cell levels). This specialization must not turn into or foster fragmentation if efforts are undertaken to keep connections between various fields, yet the danger is high. Psychological science and cognitive, affective, and social neuroscience are more fragmented than other fields of science[1]. Notably, computational simulations have shown that fragmentation is higher when there is little cross-talk with scientists taking a different view on their research topic[1]. Thus, it is essential to explicitly address and try to incorporate a different perspective in empirical research and ground research on theories from the perspective of the other community. This "other's perspective" theoretical or conceptual grounding or cross-fertilization will be essential to overcome further fragmentation in cognitive brain sciences. For the field of cognitive, affective, and social neuroscience, this means connecting more to theoretical frameworks offered by psychological science (e.g., from the field of general and experimental psychology and other psychological disciplines). Vice, versa it would be necessary for cognitive neuroscience researchers to think more "outside the box" and explicitly state implications for cognitive, affective, and social neuroscience. This will reflect an important step to create more overarching theories of "mental processes," i.e., theories that incorporate the psychological science perspective and the neuroscience perspective. This will be central because a lack of shared fundamentals (theories) is a concrete obstacle to collective progress[2] and especially overarching theories are relevant to help researchers find optimal methods to test particular hypotheses and help other researchers to connect to their research. "Unified theories of cognition" are necessary to overcome cataloging of experimental effects[3]. This is also essential since a "theory crisis" is evident[4]. It refers to the problem that there is too little theoretical progress in psychology because psychological theories are often formulated too vaguely or abstract[4]. Therefore, theory crisis in psychology is an obstacle for other research disciplines to connect to these theories and help to develop these. Addressing theory crisis by various means as described elsewhere[5] will therefore one be an essential element to overcome the disconnect

between psychological science and neuroscience. However, as outlined at the end of this article also publishers can contribute to overcome fragmentation of disciplines and associated problems.

## Effects of disconnection on reproducibility

When contemplating the necessity to build stronger connections between psychological science and various neuroscientific disciplines, aspects of replicability in science must also be considered. Paradoxically, the "replication crisis" may provide a chance to re-direct empirical research into a direction of theory-driven and theory-informative empirical research. There are many causes for the replication crisis, such as underpowered studies, publication bias, problems with the application of statistical procedures—and imprecise theories[6]. Efforts to overcome the replication crisis are of utmost importance[7]. Current efforts to overcome the replication crisis and to reinstate/corroborate trust in meaningful results of empirical research, especially the "#Manylabs" efforts in various psychological science and neuroscience fields[8–10], are essential but likely not sufficient. The reason is that from a scientific-theoretical perspective, at least two facets of replicability can be distinguished[11] and both of these are important for the reproducibility of a finding and its explanatory value[12]. Specifically, direct replications have to be distinguished from conceptual replications. The former is, for example, addressed in the #Manylabs efforts, in which experiments are repeated in the same way as they have initially been conducted, independently in various labs.

Conceptual replications, however, are quite often neglected, possibly because this form of replication is more risky and complicated to accomplish[11]. Conceptual replication means that a previous research idea/finding/hypothesis is tested using different methods. Previously, it has been stated that the goal of conceptual replications is to produce "understanding" rather than "facts"[11]. Conceptual replications require an understanding of the concepts faithfully embedded in the theory. For direct replications, an understanding of the theoretical concept is advantageous but not necessary since the repetition of previous work in exactly the same way (i.e., direct replications) does not require conceptual abstraction[11]. A fundamental requirement for conceptual replicability is that the conducted research is connected to a theoretical background in scientific publications on empirical research. It is not only the different method which is important and makes a conceptual replication[7,13], but that the different methodological procedure confers a theoretically meaningful change and possibly an extension of the previous findings. Obviously, this necessitates that a finding is accessible to researchers from different fields each having an own focus or research concept. Only work that is highly accessible to the respective other community will increase the likelihood of conceptual replications of a finding. Therefore, disconnect between research communities and problems to make findings accessible for the respective other community may impede efforts to increase reproducibility in science.

As mentioned above, there are many different "grain sizes" or levels in consideration of mental processes (from the single cell to circuitry to systems) within neuroscience. Therefore, conceptual replications can mean repeating other's work focussing on different levels of neural processes. Apparently, there is thus a very thin line between what constitutes a conceptual replication and what can be regarded to have true conceptual-theoretical novelty. As described in other's work[12], failure in direct replication does not necessarily mean that the previous result was incorrect. Slight differences in the methodology during direct replications (because these were thought to be irrelevant) can produce different findings[12] (e.g. when adapting experimental procedures from cognitive psychological/behavioral work to fit the

requirements of a specific neuroscience method). Failures of (direct) replications can, therefore, mean that specific boundary conditions of an effect are not understood in detail and have not sufficiently been considered in theoretical frameworks thus far. However, such aspects can only be correctly evaluated and interpreted if there is sufficient cross-talk and little disconnect between disciplines.

## Potential solutions for disconnect

The above sections outline why theory-guided research is essential for inter-connected topics in psychological science and neuroscience. A better connection between these communities on a common ground of theories and concepts may foster progress in science and is relevant for aspects of reproducibility. The question, however, is how can the disconnection between psychological science and neuroscience be resolved?

At this point, I think that scientific publishing is a brick that should take a central position. It is crucial to provide empirical researchers with more publishing space and an explicit forum when writing papers to make their research accessible to the broadest possible audience of researchers and to be able to connect their research to theories in another field. Many journals, especially when not "online-only," impose strict word limits for empirical research papers primarily due to economic considerations. This can make it hard for researchers to present an (extensive) theoretical background/motivation for their study and discuss their research findings in breadth. Researchers often try to present parts of their work in the supplemental material, but then the risks are high that aspects of their research will become unnoticed or misunderstood. Especially for "online journals," I think there can and should be more space to discuss possible theoretical implications, possibly in separate and dedicated sections of the paper, which would provide room for more speculative 'out-of-the-box' thinking. When doing so, it would be essential to guide such 'out-of-the-box' thinking from the publisher's site and ensure high quality of such content. This may be done by providing an organizational scheme of how possible connections between different research fields or empiry and theory can be presented. Here, quality criteria for theories may provide a framework at which level connections may be built. Out of several criteria for a good scientific theory discussed elsewhere[14], the quality features of "precision" and "generality" in particular, may be possible organizational dimensions to frame 'out-of-the-box' thinking in a meaningful way. Precision is concerned with how concepts are defined and can be operationalized. These aspects are thus crucial to building connections on a methodological level and how methods, which differ between research fields and often focus on different levels of inspection, can converge and foster conceptual replications. Authors should be encouraged to explain how researchers from other fields can connect to their work on the "precision dimension." Considering conceptual replications, the "generality" dimension is essential as this aspect relates to whether observed findings and the concept being addressed may be related and thus relevant to other fields of research. Authors should be encouraged to speculate about details on possible conceptual implications. Such a framing for creative, theory-connecting along the abovementioned dimensions will be a significant advance beyond current keyword indexing because it will provide a theoretically connectable and semantically-enriched presentation. Crucially, this may foster a theoretically meaningful mapping of results across studies that may help the scientific community explore and find conceptual bridges between research fields and levels of inspection in research. This may also better inform data science procedures such as "knowledge discovery in databases (KDD)" as a form of machine learning in semantic data[15], which may help derive more abstract insights from data and may thus also help to overcome the discussed interfacing problems between scientific communities.

One possible instrument to do this is via "connection boxes." These may have the making of a "significance statement" and may preferably be placed at the end of a paper because this box contains very condensed information of the work's essential conceptual and methodological considerations. The content of this connection box should be subject to the standard peer-review process. The text within a connection box may contain two separate subheadings oriented on the "precision dimension" and "generality dimension", as outlined above: First, the authors may state explicitly and in non-jargon terms how the concepts in the study are defined and how these were operationalized ("precision dimension"). It will also be necessary to explicitly state the theoretical framework motivating the study, including references to that theoretical framework. All this can, for example, be done in a few bullet-point sentences. The second subheading should briefly present the "generality" dimension. Here, authors should be allowed to formulate 'out-of-the-box' ideas of how researchers from other disciplines can connect to the present work. Alternatively, the connection box may also be presented graphically or as a combination of text and artwork.

However, regardless of the particular organizational structure and format, all the discussed aspects require efforts—particularly from the authors. Therefore, I think it may be essential to incentivize authors to provide the best possible content in the connection box. This may, for example, be achieved using a "badge system" (standard, silver, gold). The achieved "badge" can easily be depicted in the paper and may be a graphical element of the connection box. At a standard level (required for the acceptance of the paper), only general statements are required for which theoretical framework the results have implications and how other fields may connect to the presented finding. At higher levels, more in-depth elaborations are necessary. Evaluating which badge level has been achieved may be interactive between the editor handling the paper and the reviewers. This will also be important because it is hard to set clear criteria for levels, and fields of research may have different views on that. The evaluation's outcome should be fed back to the authors, and authors should be given a chance to enhance their presentation and conceptual depth of their connection box based on the feedback. Doing so, a fair and constructive process of perspective taking may be motivated that may help reducing disconnect in science and contribute to the (further) development of more holistic theories of 'mind' and the brain.

## Concluding remarks

Regardless of its length, possible format, and implementation in an article for out-of-the-box thinking on theoretical implications, authors would be given the tools by publishers to provide theoretical background in a concise way that it is accessible to other researchers in related fields who may have a different perspective on the research question. By actively encouraging researchers and by providing a dedicated forum or space in research papers, publishers can play their part in strengthening connections between fields of research or, in some cases, establishing new ones. By doing so, publishing is transformed as it becomes a more active and supporting process in the formation of scientific theory and progress that goes beyond documentation of the scientific discourse.

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

## Acknowledgements

I thank my many colleagues for stimulating discussions related to the presented topic and the two reviewers for their comments. This work was supported by Grants from the Deutsche Forschungsgemeinschaft, SFB 940, SFB TRR 265 and FOR 2698.

## Author contributions

CB: conceptualization and writing.

## Funding

## Competing interests

The author declares no competing interests.
