## [Peer Review File · Communications Biology]

Disconnected psychology and neuroscience – implications for scientific progress, replicability and the role of publishingReviewers' comments:

Reviewer #1 (Remarks to the Author):

Even though neuroscience and psychology (frequently) address the same research questions, mind/cognition research are fragmented. According to the author, this issue impedes not only scientific progress but also the replicability of research. The manuscript focuses on an "interface problem" between psychology and neuroscience and proposes important steps toward its solution at the institutional level.

This is well written and interesting manuscript. Its topic is very important and "hot" for a broad community of brain and cognition researchers. The claims drawn by the author are sound and could be a significant stimulus for future integrative efforts in the field(s). For me, publishers and peers involved in the publishing process should implement, or at least consider, the author's voice.

I have no major objections to the manuscript. However, I think that the author should provide more detail of his proposal about including more theoretical background in publications (see below). Moreover, I list some (minor) things for considering by the author.

L121-3: "Conceptual replications, however, are quite often neglected, possibly because this form of replication is more risky and complicated to accomplish". This is interesting observation (additionally supported with a paper by Schmidt). I think, however, that the issue is more tricky. According to the author (L123-5): <<Conceptual replication means that a previous research idea/finding/hypothesis is tested using different methods. The goal of conceptual replications is to produce "understanding" rather than "facts">>. In the Zwaan et al.'s paper (2018, BBS, p. 3-4) a conceptual replication is considered as "a study where there are changes to the original procedures that might make a difference with regard to the observed effect size. Conceptual replications span a range from having one theoretically meaningful change with regard to the original experiment (e.g., a different dependent measure) to having multiple changes (Lebel et al. 2017). (...) What such a study does, in effect, is test an extension of the theory to a new context (because there are different auxiliary hypotheses involved in the operationalization of the key variables)". Thus, I think a conceptual replication reaches beyond just "using different methods". Moreover, I do not understand the author's claim that conceptual replications produce <<"understanding" rather than "facts">> (L124-5). I also do not follow with the note about Popper's falsifications (L125-7). I think the author should make more explicit why this appears in the context of conceptual replications. Maybe the reason is that a conceptual replication should be faithfully embedded in the theory. But this also applies to direct replications – see Zwaan et al.'s paper (p. 3).

I have one more observation regarding replications: for me, the problem with conceptual replications in integrative efforts in mind/cognition research also lies in the fact that many conceptual replications are "branded" in the literature as original or completely novel research. In other words: we have a lot of conceptual replications in the field(s), but their "nature" is not considered adequately, impeding theoretical progress.

L100-4: Generally, the problem of fragmentation of brain/cognition research and the need for building unificatory accounts are well-described in recent literature. I think mentioning this fact could be beneficial for readers. The author could check e.g., Núñez, R., Allen, M., Gao, R., Rigoli, C. M., Relaford-Doyle, J., & Semenuks, A. (2019). What happened to cognitive science?. *Nature Human Behaviour*, 3(8), 782-791. <https://doi.org/10.1038/s41562-019-0626-2> and Miłkowski, M., Hohol, M. (2020). Explanations in cognitive science: Unication versus pluralism. *Synthese*. Online first. <https://doi.org/10.1007/s11229-020-02777-y>. Also, theory crisis in psychological science is also the subject of recent discussions: Eronen, M. I., & Bringmann, L. F. (2021). The theory crisis in psychology: How to move forward. *Perspectives on Psychological Science*, <https://doi.org/10.1177/1745691620970586>; Oberauer, K., Lewandowsky, S. (2019). Addressing the theory crisis in psychology. *Psychonomic Bulletin & Review*, 26(5), 1596-1618. <https://doi.org/10.3758/s13423-019-01645-2>. At least briefly mentioning these discussions would be beneficial for readers and make the manuscript more up to date.

My crucial impression is about steps toward the solution of the fragmentation issue proposed by

the author:

L152-4 & 169-172: <<Especially for "online journals," I think there can and should be more space to discuss possible theoretical implications, possibly in separate and dedicated sections of the paper, which would provide room for more speculative 'out-of-the-box' thinking>>. I totally agree, but at the same time, I am curious about more detailed proposals regarding this matter. Should <<more speculative 'out-of-the-box' thinking>> be embedded in the introduction or rather discussion of a research paper? Maybe special "boxes" should be implemented? Or the traditional structure of a research paper should be revisited (to include additional section/s)? Possibly such "institutional" changes could facilitate searching for information about the employed theoretical framework both by human readers and artificial interfaces (e.g., machine learning driven)?

Other small things:

L36: "... what underlies behavior psychology and neuroscience". It seems, the comma between 'behavior' and 'psychology' is missing.

L102 & L204-5: Hacking is the author of an introduction to the cited edition of "The Structure...", but I think only Kuhn should be indicated in the reference as the author.

Using `,' and `,"' should be more consistent.

All the best, Mateusz Hohol

Reviewer #2 (Remarks to the Author):

The author makes a case for connecting psychology and neuroscience better than it is actually the case and he also provides an idea how publishers can contribute to this issue.

Generally speaking, I like the idea of this manuscript. I here list some ways to improve it.

1. Psychology and neuroscience are fragmented and in addition oftentimes they ignore the other discipline. I think these are two separate aspects (or issues for progress) that sometimes are confused by the author. I get that he argues that the fragmentations potentially enhance the separation but this should be made clearer.

Concerning the separation one might also take a different stance (that is actually completely ignored in this opinion paper). In fact, philosophers like D. Dennett have argued long ago that there are separate levels of analyzing the mind, eg the neural level, the behavioral level and so on. One does not need to understand say the neural level to make great progress and great research at the behavioral level and vice versa. I think the concept of inhibition that has seen a lot of controversy is a good example – inhibition in the synaptic cleft, at the level of neural responses, at the level of physiological activation patterns, and at the level of inhibiting responses shows that one does not need to connect these levels for progress. To put it bluntly, cognitive inhibition models do not need a connection to the level of neurons etc.

(I would like to add that this is not my personal opinion)

2. I of course understand the difference between exact and conceptual replications but I did not understand how the latter actually affect reproducibility. This paragraph seems to be missing some lines/sentences? Or I lost the argumentation punch line here. In any case, I had the strong feeling that the whole topic of the replication crisis is not needed for the point the authors wants to make. As I see it, the main point is to overcome the separation of psychology and neuroscience – and this is only mildly related to the replication crisis and the role conceptual replication have there, no?

3. The idea that publishers can force the connection is good – at least in theory (I think journal space is so expensive in print formats at least). But, if and only if the publishers were 'on board' I would suggest to slightly change the idea the author has here. Why not have an extra box/part of the paper that is devoted to the 'other' field. And is also reviewed by a reviewer from the other field? I don't think this should be a one-way thing. Neuroscientist in my experience need sometimes a bit more theory, so they should be 'forced' to think and frame their research in overarching theories from psychology or at least state what they contribute. At the same time psychology papers might have the very same section in which they are 'forced' to discuss what their theory/idea means at the neural or physiological level – which brain areas might be involved,

which processes might underlie the observable behavior and so on. Only if both communities try to connect this can be fruitful in the middle/long run. I would also recommend that neuroscience papers might have a box/extra section that explain in simple ways the methods the use. Whether this is realistic, I don't know. But another suggestion could be that at least some journals specialize in this approach and these journals should force authors to connect their research to other fields.

Reviewer 1:

Even though neuroscience and psychology (frequently) address the same research questions, mind/cognition research are fragmented. According to the author, this issue impedes not only scientific progress but also the replicability of research. The manuscript focuses on an “interface problem” between psychology and neuroscience and proposes important steps toward its solution at the institutional level.

This is well written and interesting manuscript. Its topic is very important and “hot” for a broad community of brain and cognition researchers. The claims drawn by the author are sound and could be a significant stimulus for future integrative efforts in the field(s). For me, publishers and peers involved in the publishing process should implement, or at least consider, the author’s voice. I have no major objections to the manuscript. However, I think that the author should provide more detail of his proposal about including more theoretical background in publications (see below). Moreover, I list some (minor) things for considering by the author.

I thank this reviewer for the positive evaluation of the work.

L121-3: “Conceptual replications, however, are quite often neglected, possibly because this form of replication is more risky and complicated to accomplish”. This is interesting observation (additionally supported with a paper by Schmidt). I think, however, that the issue is more tricky. According to the author (L123-5): <<Conceptual replication means that a previous research idea/finding/hypothesis is tested using different methods. The goal of conceptual replications is to produce “understanding” rather than “facts”>>. In the Zwaan et al.’s paper (2018, BBS, p. 3-4) a conceptual replication is considered as “a study where there are changes to the original procedures that might make a difference with regard to the observed effect size. Conceptual replications span a range from having one theoretically meaningful change with regard to the original experiment (e.g., a different dependent measure) to having multiple changes (Lebel et al. 2017). (...) What such a study does, in effect, is test an extension of the theory to a new context (because there are different auxiliary hypotheses involved in the operationalization of the key variables)”. Thus, I think a conceptual replication reaches beyond just “using different methods”.

Thank you for this comment. This is true and it may have been misunderstandable. This section was revised accordingly.

Moreover, I do not understand the author’s claim that conceptual replications produce <<“understanding” rather than “facts”>> (L124-5). I also do not follow with the note about Popper’s falsifications (L125-7). I think the author should make more explicit why this appears in the context of conceptual replications. Maybe the reason is that a conceptual replication should be faithfully embedded in the theory. But this also applies to direct replications – see Zwaan et al.’s paper (p. 3).

Thank you for your comments. I have clarified this aspect in the text in connection to the comment above. In fact, conceptual replications require an understanding of the concepts faithfully embedded in the theory. For direct replications, however, an understanding of theoretical concepts is advantages, but not necessary since the repetition of previous work in exactly the same way as it has been conducted before (i.e. direct replications) does not require conceptual abstraction.

I have one more observation regarding replications: for me, the problem with conceptual replications in integrative efforts in mind/cognition research also lies in the fact that many conceptual replications are “branded” in the literature as original or completely novel research. In other words: we have a lot of conceptual replications in the field(s), but their “nature” is not considered adequately, impeding theoretical progress.

I only briefly added this since this is a topic on its own. Nevertheless, it should be mentioned.

*L100-4: Generally, the problem of fragmentation of brain/cognition research and the need for building unificatory accounts are well-described in recent literature. I think mentioning this fact could be beneficial for readers. The author could check e.g., Núñez, R., Allen, M., Gao, R., Rigoli, C. M., Relaford-Doyle, J., & Semenuks, A. (2019). What happened to cognitive science?. *Nature Human Behaviour*, 3(8), 782-791. <https://doi.org/10.1038/s41562-019-0626-2> and Milkowski, M.,*

Hohol, M. (2020). Explanations in cognitive science: Unication versus pluralism. Synthese. Online first. <https://doi.org/10.1007/s11229-020-02777-y>. Also, theory crisis in psychological science is also the subject of recent discussions: Eronen, M. I., & Bringmann, L. F. (2021). The theory crisis in psychology: How to move forward. Perspectives on Psychological Science, <https://doi.org/10.1177/1745691620970586>; Oberauer, K., Lewandowsky, S. (2019). Addressing the theory crisis in psychology. Psychonomic Bulletin & Review, 26(5), 1596–1618. <https://doi.org/10.3758/s13423-019-01645-2>. At least briefly mentioning these discussions would be beneficial for readers and make the manuscript more up to date.

Thank you for these valuable comments. I included these thoughts into the section on fragmentation in science.

My crucial impression is about steps toward the solution of the fragmentation issue proposed by the author:

L152-4 & 169-172: <<Especially for "online journals," I think there can and should be more space to discuss possible theoretical implications, possibly in separate and dedicated sections of the paper, which would provide room for more speculative 'out-of-the-box' thinking>>. I totally agree, but at the same time, I am curious about more detailed proposals regarding this matter. Should <<more speculative 'out-of-the-box' thinking>> be embedded in the introduction or rather discussion of a research paper? Maybe special "boxes" should be implemented? Or the traditional structure of a research paper should be revisited (to include additional section/s)? Possibly such "institutional" changes could facilitate searching for information about the employed theoretical framework both by human readers and artificial interfaces (e.g., machine learning driven)?

Thank you. I thought on the topic of a "connection box" before, but was a bit hesitant on this.

However, since also the second reviewer suggests this, I added a few first thought on how this may be implemented.

Other small things:

L36: "... what underlies behavior psychology and neuroscience". It seems, the comma between 'behavior' and 'psychology' is missing.

Added.

L102 & L204-5: Hacking is the author of an introduction to the cited edition of "The Structure...", but I think only Kuhn should be indicated in the reference as the author.

I added the original reference to Kuhn and replaced the one by Hacking and Kuhn.

Using ',' and ',' should be more consistent.

Done.

All the best, Mateusz Hohol

Thank you.

Reviewer #2 (Remarks to the Author):

The author makes a case for connecting psychology and neuroscience better than it is actually the case and he also provides an idea how publishers can contribute to this issue.

Generally speaking, I like the idea of this manuscript. I here list some ways to improve it.

1. Psychology and neuroscience are fragmented and in addition oftentimes they ignore the other discipline. I think these are two separate aspects (or issues for progress) that sometimes are confused by the author. I get that he argues that the fragmentations potentially enhance the separation but this should be made clearer.

Thank you for pointing this. I addressed this aspect at the end of the second section (fragmentation) in combination with aspects of the first reviewer.

Concerning the separation one might also take a different stance (that is actually completely ignored in this opinion paper). In fact, philosophers like D. Dennett have argued long ago that there are separate levels of analyzing the mind, eg the neural level, the behavioral level and so on. One does not need to understand say the neural level to make great progress and great research at the behavioral level and vice versa. I think the concept of inhibition that has seen a lot of controversy is a good example – inhibition in the synaptic cleft, at the level of neural responses, at the level of physiological activation patterns, and at the level of inhibiting responses shows that one does not need to connect these levels for progress. To put it bluntly, cognitive inhibition models do not need a connection to the level of neurons etc. (I would like to add that this is not my personal opinion)

Thank you very much for this. I have added the valuable thoughts to the manuscript.

2. I of course understand the difference between exact and conceptual replications but I did not understand how the latter actually affect reproducibility. This paragraph seems to be missing some lines/sentences? Or I lost the argumentation punch line here. In any case, I had the strong feeling that the whole topic of the replication crisis is not needed for the point the authors wants to make. As I see it, the main point is to overcome the separation of psychology and neuroscience – and this is only mildly related to the replication crisis and the role conceptual replication have there, no?

Thank you for pointing this out. Indeed, this may not have become clear enough and I revised the section of the paper.

3. The idea that publishers can force the connection is good – at least in theory (I think journal space is so expensive in print formats at least). But, if and only if the publishers were ‘on board’ I would suggest to slightly change the idea the author has here. Why not have an extra box/part of the paper that is devoted to the ‘other’ field. And is also reviewed by a reviewer from the other field? I don’t think this should be a one-way thing. Neuroscientist in my experience need sometimes a bit more theory, so they should be ‘forced’ to think and frame their research in overarching theories from psychology or at least state what they contribute. At the same time psychology papers might have the very same section in which they are ‘forced’ to discuss what their theory/idea means at the neural or physiological level – which brain areas might be involved, which processes might underlie the observable behavior and so on. Only if both communities try to connect this can be fruitful in the middle/long run. I would also recommend that neuroscience papers might have a box/extra section that explain in simple ways the methods the use. Whether this is realistic, I don’t know. But another suggestion could be that at least some journals specialize in this approach and these journals should force authors to connect their research to other fields.

Thank you. I thought on the topic of a “connection box” before, but was a bit hesitant on this.

However, since also the first reviewer suggests this, I added a few first thought on how this may be implemented.

There were no reviewer comments left.